# DOC export is exceeded by C fixation in May Creek: A late-successional watershed of the Copper River Basin, Alaska

**Patrick L. Tomco**[1]* , **Rommel C. Zulueta**[2], **Leland C. Miller**[3], **Phoebe A. Zito**[4], **Robert W. Campbell**[5], **Jeffrey M. Welker**[3,6]

**1** Department of Chemistry, University of Alaska Anchorage, Anchorage, Alaska, United States of America, **2** National Ecological Observatory Network, Inc., Boulder, Colorado, United States of America, **3** Department of Biological Sciences, University of Alaska Anchorage, Anchorage, Alaska, United States of America, **4** Department of Chemistry, University of New Orleans, New Orleans, Louisiana, United States of America, **5** Prince William Sound Science Center, Cordova, Alaska, United States of America, **6** Ecology and Genetics Research Unit, University of Oulu, Oulu, Finland

* pltomco@alaska.edu

**Data Availability Statement:** All relevant data are within the paper and its Supporting Information files.

## Abstract

Understanding the entirety of basin-scale C cycling (DOC fluxes and $CO_2$ exchanges) are central to a holistic perspective of boreal forest biogeochemistry today. Shifts in the timing and magnitude of dissolved organic carbon (DOC) delivery in streams and eventually into oceans can be expected, while simultaneously $CO_2$ emission may exceed $CO_2$ fixation, leading to forests becoming stronger $CO_2$ sources than sinks amplifying rising trace gases in the atmosphere. At May Creek, a representative late-successional boreal forest watershed at the headwaters of the Copper River Basin, Alaska, we quantified the seasonality of DOC flux and landscape-scale $CO_2$ exchange (eddy covariance) over two seasonal cycles. We deployed *in situ* fDOM and conductivity sensors, performed campaign sampling for water quality (DOC and water isotopes), and used fluorescence spectroscopy to ascertain DOC character. Simultaneously, we quantified net $CO_2$ exchange using a 100 ft eddy covariance tower. Results indicate DOC exports were pulse-driven and mediated by precipitation events. Both frequency and magnitude of pulse-driven DOC events diminished as the seasonal thaw depth deepened, with inputs from terrestrial sources becoming major contributors to the DOC pool with decreasing snowmelt contribution to the hydrograph. A three-component parallel factorial analysis (PARAFAC) model indicated DOC liberated in late-season may be bioavailable (tyrosine-like). Combining Net Ecosystem Exchange (NEE) measurements indicate that the May Creek watershed fixes 142–220 g C m$^{-2}$ yr$^{-1}$ and only 0.40–0.57 g C m$^{-2}$ yr$^{-1}$ is leached out as DOC. Thus, the May Creek watershed and similar mature spruce forest dominated watersheds in the Copper River Basin are currently large ecosystem C sinks and exceeding C conservative. An understanding of DOC fluxes from Gulf of Alaska watersheds is important for characterizing future climate change-induced seasonal shifts.

**Funding:** Funding for this project was provided to RC and JW by NASA (grant# NNX10AU07G). https://www.nssc.nasa.gov/grantstatus. The funders had no role in study design, data collection and analysis, decision to publish, or preparation of the manuscript.

**Competing interests:** The authors have declared that no competing interests exist.

# Introduction

The boreal forest is an important biome because of its large role in the terrestrial carbon (C) cycle [1]. The boreal forest fixes up to 20% of the earth's C [2], almost entirely controls the seasonal fluctuations in the northern hemispheric $CO_2$ concentration [3], and retains over 30% of the planet's total soil C, much of it in the form of permafrost [4, 5]. This pool of soil C is twice that of the contemporary atmosphere [6] and may have a series of fates as the global climate changes. The boreal forest C cycle is comprised of both gaseous fixation and losses (i.e. $CO_2$ and $CH_4$) and dissolved fluxes from soils into streams, rivers and long-distance oceanic sinks as DOC [7, 8]. In order to fully appreciate the relative magnitudes of trace gas C and DOC we require studies that are comprehensive in boreal forest C measurements and include detailed studies of the nature of DOC, as the quality of DOC may have important consequences for stream, river and near-shore marine food webs [9].

The boreal forest region of coastal Gulf of Alaska (GoA) has received much attention recently due to observations of massive glacier melt and permafrost thaw as manifested by rapid and accelerating climate change [10, 11]. Climate models predict up to a 40% increase in runoff from Alaska rivers by 2050 due to glacial melt [12], and over the coming decades an increase in river discharge is predicted to taper as glacial melt and permafrost thaw rates near completion [11]. There is however, considerable uncertainty over how this region and its glacial-soil-aquatic-marine system will change in the next 40 years. Some predict new habitats for salmon as the headwater streams become increasingly suitable for spawning as deglaciated landscapes undergo succession [13, 14], while conversely and/or simultaneously, permafrost and decreased river flows may lead to the loss of habitat as freshwater sources dry seasonally or permanently.

Changes in the source and magnitude of freshwater discharge in headwater streams are likely to alter downstream nutrient biogeochemistry in the coastal Gulf of Alaska (GoA). Recent studies have shown that glacial melt-derived nutrients fuel heterotrophic growth in the nearshore GOA [15–18]. Nutrient flux in catchments with significant glacial coverage is dominated by melting processes of these glaciers and the associated DOC pulse that is released from previously cryogenic- C, which is mostly lysed, readily biodegradable microbial cellular matter [16]. At locations where glaciers have recently receded, vegetation is characterized by nitrogen-fixing plants *Dryas spp.* and *Shepherdia spp.* and have little/no organic horizon to adsorb/sequester meltwater C. Over time as melting nears completion, C flux in these sites will less resemble glacial melt and shift to being dominated by atmospherically-fixed C with DOC character more resembling the new vegetation from spruce, birch, cottonwood, and willow.

Terrestrial vegetation contributes to stream water DOC through litter fall and/or exudation, and also generates soil organic carbon (SOC), which in turn can modify throughflow and groundwater storage characteristics. Litter inputs relevant to these processes consist of coniferous needles, twigs, stems, and logs, and waxy ericaceous leaves [19]. The understory is covered with thick mats of *Sphagnum* that have been shown to significantly contribute to DOC in the form of exudates [20]. Vegetation-derived SOC accumulates in the mineral layer as vegetation succession proceeds. Near-surface SOC in the organic horizon accumulates at faster initial rate, but reaches a maximum quickly, and is readily mobilized by disturbances, i.e. fire [21].

Northern ecosystems have carbon cycles that can be biogeochemically complex, whereby aqueous and gaseous C fluxes can determine the fate of land surface exchanges with the surrounding atmosphere and adjoining rivers and subsequent marine systems. Recently, for instance, [22] have indicated that the boreal system of Eurasia is contributing over 1.75 Tg C to the Arctic Ocean via DOC efflux and that over the course of the summer, the source of this C becomes progressively dominated by ancient permafrost C. The Yukon River, like the Ob,

Yenesi, Ingidin, and Mackenzie is also a major catchment which transports C from interior Alaska to the Bering Sea [23], while smaller watersheds in south-central and southeastern Alaska contribute to the C cycle of near shore marine systems in Prince William Sound and the Gulf of Alaska, running North-South. However, while these DOC studies have quantified one form of C export, they are seldom accompanied by measurements which quantify the net $CO_2$ exchange of the same ecosystem, nor the standing pool of C in trees [24, 25], making it difficult to appreciate the relative role of DOC export as a fraction of C sequestration, a more holistic assessment of watershed C cycling.

Fluorescence spectroscopy is a tool that is commonly utilized to characterize the chemical constituency of dissolved organic matter in freshwater systems. Recent efforts have been directed at characterizing other Alaskan freshwater aquatic ecosystems in the Tongass [15, 16] and Yukon [26–28], among others in the boreal [29–31]. To date, no studies have attempted to classify seasonal fluorescence characteristics anywhere in the Copper River Basin or the Wrangell-St. Elias region of Alaska.

At May Creek, a representative climax spruce forest at the headwaters of the Copper River, we examined the physical, biogeochemical, and hydrologic processes that control dissolved organic C export to the river and relate the export to $CO_2$ flux values generated via eddy covariance [25]. A combination of expedition sampling and deployed *in-situ* sensors was used to observe dissolved components, and an eddy covariance flux tower was installed at the field site to observe Net Ecosystem Exchange (NEE) flux values over the same timeframe. The goal of this study was to determine the seasonal and inter-annual variability in the timing and magnitude of dissolved organic C export, determine how the chemical characteristics of dissolved C fluctuated seasonally and between water source types, and to relate DOC export to C fixation. Understanding these terrestrial-aquatic interactions is an important step in our depiction of these processes, and is useful for furthering upscaling efforts to determine basin-wide nutrient fluxes.

## Materials and methods

### Study site

The Copper River Basin watershed contains landscape types which have been previously delineated [17] as glacial, proglacial lake (early successional), non-glacial Boreal montane, and boreal lowland (late successional). To employ a space-for-time assumption for successional processes, we selected a non-glacial montane watershed for our study site. We hypothesized that atmospherically-fixed C accumulates rapidly in the standing biomass and accounts for the majority of the total C budget in headwater streams, as evidenced by annual Net Ecosystem Exchange (NEE) and DOC export masses.

May Creek is a climax spruce forested watershed (31.5 km$^2$) at the headwaters of the Copper River (**Fig 1**). It is underlain with discontinuous permafrost at depth of 50 cm. The basin ranges in elevation from 450–1400 m. The vegetation consists primarily of mixed white/black spruce, cottonwoods, peat, lichens, mosses, and willow shrubs (*Salix*), and the forest floor is overlain with a thick organic mat up to ~0.3 m. The soil consists of an organic layer underlain by clay sediment. The site is accessible only by fixed-wing aircraft from McCarthy, the nearest road-accessible town. To account for the spectrum of water types in the region, our sampling sites (**Fig 1**) included the following: 1) May Creek (MC, 61.3485667˚, -142.6967833˚), 2) a nearby subsurface-fed spring within the watershed (MC Spring, 61.3493000˚, -142.6926333˚), 3) Young Creek (61.3497500˚, -142.7266000˚), a neighboring creek that drains a larger (289 km$^2$) non-glacial higher elevation (450–2,609 m,) watershed that includes both alpine grassland tundra and spruce forest at lower elevations, and 4) MC Wetland (61.3476333˚,

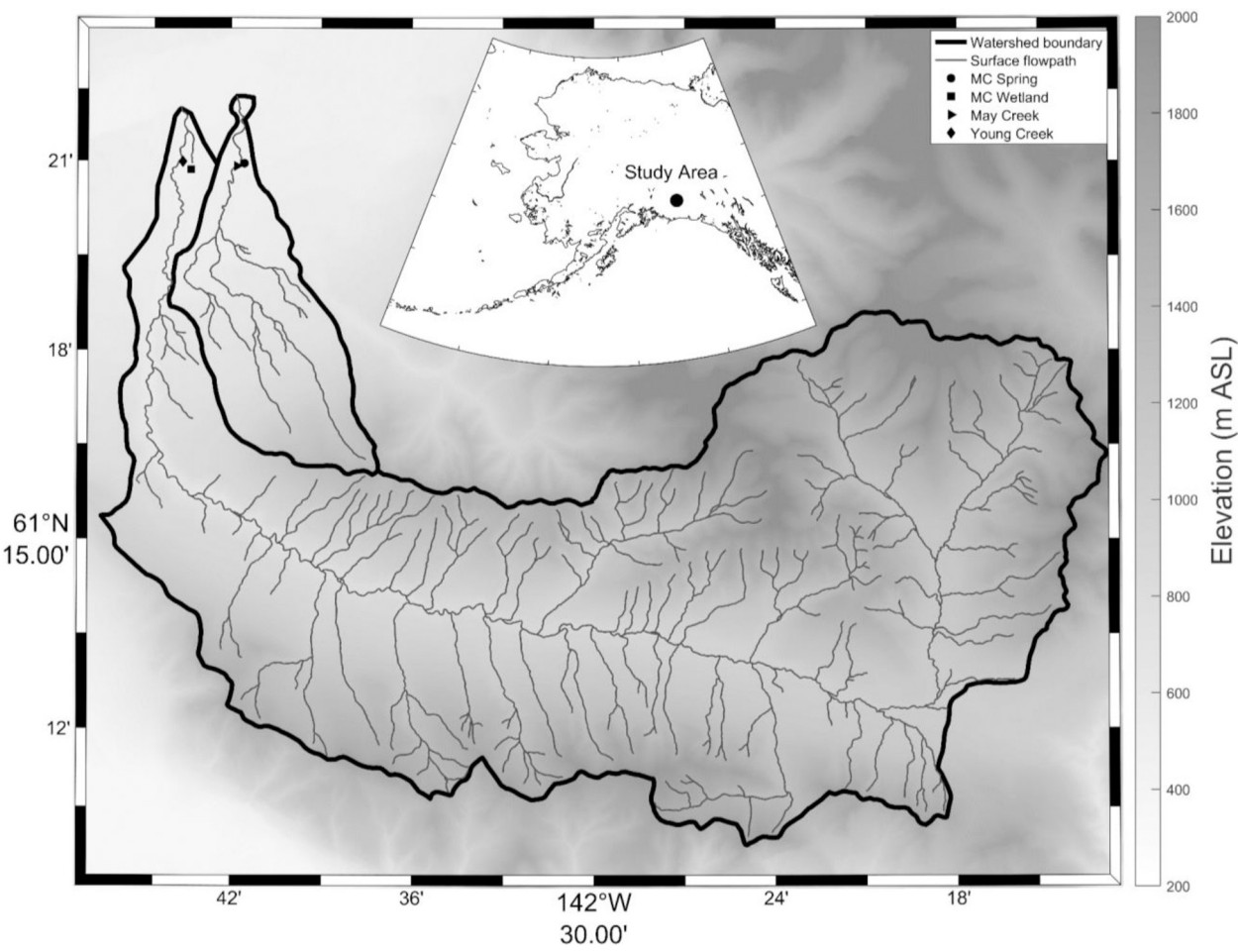

**Fig 1. Site map and sampling locations.**

-142.7220833˚), a lowland site that receives a combination of subsurface thaw and terrestrial throughflow.

## Collection and analysis of water samples

Individual water samples were collected during the 2012 and 2013 field seasons at approximately biweekly intervals from May to September. Each sample was collected in an acid-rinsed 1 L borosilicate amber vial, triple rinsed with river water, submerged until filled, and capped with no headspace. Samples were filtered by passing water through a 0.45 um Acrodisk® syringe-driven unit. Any leachable DOC on the filter was flushed by passing 100 mL DI water through the filter and discarding this filtrate prior to collecting in sample vials. Samples were filtered into 40-mL VOC TraceClean amber vials for DOC analysis, 2-mL autosampler vials (National Scientific) for isotope analysis, and acid-rinsed PTFE bottles for fluorescence analysis, each of which were rinsed once with filtered stream water. Samples were transported to the laboratory on ice. DOC was analyzed using a Shimadzu TOC-VCHN analyzer at the University of Alaska Anchorage ASET laboratory. Water isotopic values ($\delta^{18}$O SMOW) were determined at the UAA Stable Isotope Laboratory with a Picarro WS-CRDS system utilizing a CTC Analytics PAL autosampler to deliver a liquid water sample via injection into a vaporization chamber. The vaporized water sample is fed into a closed cavity for analysis, during which a

discrete quantity of laser light is introduced into the cavity and the absorption curve of the light is used to determine the isotopic ratio ($^{18}O/^{16}O$ and $^{2}H/^{1}H$) of the water sample.

## Hydrological and in-situ sensor data

A Hach FH950 portable velocity meter and a top-end wading rod were used to estimate stream discharge during sampling campaigns at a reach of 200 cm. The stream was partitioned into 10-cm intervals and discharge calculated with two-point velocity measurements (0.2 and 0.8 depth) for stage (h) > 30 cm and a 1-point (0.6 depth) measurement for h< 30 cm. An Onset HOBO U20 pressure transducer was deployed *in situ* and stage calculated at 5-min intervals using atmospheric pressure readings taken at the eddy covariance tower. The stage-discharge relationship was log-log transformed and linearly regressed to yield a ratings curve. Discharge (Q) was determined by regressing stage to discharge.

Specific conductance was measured at 5-minute intervals with an Onset HOBO U24 conductivity data logger. The unit was calibrated several times throughout the season with a 1-point calibration standard. Fluorescent dissolved organic matter (fDOM) is the fluorescent component of chromophoric dissolved organic matter (CDOM), a component of the DOC pool that can be monitored in real time using *in-situ* devices. We took fDOM readings in May Creek at 1-minute intervals with a Cyclops7 CDOM sensor (Turner Designs, Sunnyvale, California) connected to a Campbell data logger. The unit was suspended on a rod at 30 cm depth. All fDOM data (collected in mV) correspond to fluorescence at excitation/emission wavelengths 325/470 nm calibrated to ppb PTSA (1,3,6,8-pyrenetetrasulfonic acid) with serially-diluted standards, per manufacturer recommendation. Data were time-averaged to 5-minute intervals to synchronize with the U24 and U20 sensor data. All units deployed in May Creek were checked for fouling and debris at each sampling campaign and cleared as necessary.

## Spectrofluorometric data

Fluorescence readings were taken on an Aqualog spectrofluorometer (Horiba Scientific, Edison, NJ) equipped with an internal UV spectrophotometer and analyzed according to the procedures of [32]. Briefly, acid-rinsed 1-cm cuvettes were triple rinsed with Millipure water and then filtered sample prior to loading in the instrument. Excitation emission matrices (EEMs) were generated over excitation 240–450 nm (5 nm increments) and emission 300–600 nm (2 nm increments). Millipure water blanks were collected every 5 samples in addition to a daily Raman blank that was used for normalization under the water Raman curve to adjust for instrument-specific bias. Concentrated samples (>5 ppm DOC) were diluted 10x and corrected by dilution factor. Integration time was 0.1 seconds. All samples were collected in S/R ratio mode, blank subtracted, inner-filter corrected using the manufacturer's supplied software, and Raman-normalized. Method blank checks consisting of DI water transported to the field revealed no contamination.

Fluorescence Index (FI) values were calculated according to the procedures of [33]:

$$FI = \frac{I_{470}}{I_{520}}$$

For Excitation = 370 nm. FI is commonly used as a terrestrial vs. biogenic index, with lower values between 1.2–1.5 indicating presence of highly terrestrial-derived fulvic acids and higher values indicating a more microbial-derived source [34]. Humification Index (HIX) values

were calculated according to [35] as

$$HIX = \frac{\sum I_{300 \to 345}}{\sum I_{300 \to 345} + \sum I_{435 \to 480}}$$

For Excitation = 255 nm. HIX values range from 0–1, with values closer to 1 indicating a highly humified character and lower values indicating a highly proteinaceous (tyrosine or tryptophan) character [35]. Parallel factor analysis (PARAFAC) was conducted on water samples using the drEEM toolbox for Matlab (Mathworks, Natick MA) [36, 37]. A three-component model was validated by split half analysis and through visible inspection of spectral loadings [38]. The percent relative contribution of each component calculated to the sum of total fluorescence is reported for each water sample.

## Eddy covariance flux measurements

Tower-based ecosystem exchange measurements were quantified using the eddy covariance (EC) technique [39, 40] on a 30-meter-tall scaffold tower. Measurements of the three-dimensional wind and sonic temperature were done with an ultrasonic anemometer (CSAT3, Campbell Scientific, Inc., Logan, UT, USA) while high-frequency fluctuations of $CO_2$ and water vapor were done with a closed-path infrared gas analyzer (IRGA; LI-7200, LI-COR, Inc., Lincoln, NE). A 0.95 cm outer diameter and 100 cm long insulated copper intake tubing was used, with the air intake cup 15 cm behind and 1 cm below the center control volume of the sonic anemometer. The sampling flow rate through the IRGA was maintained at 15 LPM. The IRGA was calibrated twice monthly with a $CO_2$-free air and 450 ppm $CO_2$ gas standards, and a portable dew point generator (LI-610, LI-COR, Inc.). The turbulence parameters and $CO_2$ and water vapor fluctuations were sampled at 20 Hz with the raw time series data stored on a data logger (LI-7550, LI-COR, Inc.).

Measurements at the top of the tower included incoming and reflected short- and long-wave radiation (NR01, Hukseflux, Delft, Netherlands), photosynthetically active radiation (PAR; LI-190SB, LI-COR, Inc.), air temperature and relative humidity (HMP155, Vaisala, Helsinki, Finland), wind vector (034B, Met One Instruments, Grants Pass, OR), and barometric pressure (PTB101B, Vaisala). Wind profile measurements (014A, Met One Instruments) were done above the ground, while $CO_2$ and water vapor concentration profiles were measured within the canopy (AP200, Campbell Scientific, Inc.). Ground measurements included soil temperature (Type-T thermocouples, Omega Engineering, Stamford, CT, USA), ground heat flux (HFP01, Hukseflux), and soil moisture (EC-10, Decagon Devices, Pullman, WA, USA). The slow response meteorological sensors were sampled at 10-s intervals and stored as half-hourly averages. Precipitation data were obtained from the National Resource Conservation Service (NRCS) SNOTEL site located at May Creek.

Raw data processing was done using EddyPro® open source software (v5.1.1; LI-COR, Inc.) which included spike detection and removal [41], and physical thresholds for out-of-range values. Fluxes of $CO_2$, $H$, and $\lambda E$ were calculated as half-hourly block averages with a double coordinate rotation [42] and time lag compensation using a covariance maximization method. Since a closed-path IRGA was used, the gas concentrations were expressed as mixing ratios and therefore was not necessary to apply the WPL [43] corrections to compensate for air density fluctuations [44, 45]. Spectral corrections were performed for both high-pass [46] and low-pass [47] filtering effects. We used the crosswind-integrated flux footprint model of [48] to estimate the upwind sampling area. Additional quality checks on the processed fluxes were evaluated using the overall quality flags described in [49]. Calculated fluxes were u*-filtered [50, 51] with a threshold of 0.1 before gap-filling [52, 53] was performed with the Marginal

Distribution sampling method [54] using REddyProcWeb (https://www.bgc-jena.mpg.de/bgi/index.php/Services/REddyProcWeb).

## Statistical analysis

Statistical tests were performed using JMP 11.0 (SAS institute, Cary, North Carolina). Unless otherwise stated, group-wise mean comparisons were performed via Student's t-test, while regression analysis is reported as analysis of variance (ANOVA) test of significance for a linear model. Significances are noted at 95% confidence.

## Results and discussion

The Copper River basin of interior Alaska is one of the larger in all of Alaska and North America, being in the top 10 and on par with those such as the Yukon, McKenzie, Columbia, Mississippi and Missouri. It is ~ 7.3 million hectares (about the size of West Virginia and is the largest single freshwater source to the GoA, with an annual discharge of ~65 km$^3$; that is second in Alaska only to the Yukon River (11). At this scale, biogeochemical processes are thus exceedingly important while having complex hydrologic and abiotic factors that govern dissolved C export from surface and subsurface flowpaths and atmospheric C exchanges between this boreal forest and the atmosphere. Here, we discuss and present a perspective on the C cycle of this boreal forest watershed that combines dissolved organic C export with fixed C inputs.

### DOC export and water isotopes

DOC concentrations at May Creek were highest during the spring freshet when stored C in the snowpack was liberated to the neighboring flow path (**Fig 2A**). For 2012 and 2013, DOC levels were 10.1 and 19.6 mg/L on May 14$^{th}$, the first sampling date for each year. These values decreased seasonally as samples collected on August 20$^{th}$ were 4.5 and 0.94 mg/L for 2012 and 2013. For Young Creek, the 2012 and 2013 values were 6.7 and 11.2 mg/L on May 14$^{th}$ decreasing to 1.02 and 1.62 mg/L on August 20$^{th}$. MC Spring DOC values increased slowly during 2012 (*p<0.0122*) from 0.68 mg/L on May 14$^{th}$ to 0.83 mg/L on August 20$^{th}$. For 2013, MC Spring DOC did not significantly increase seasonally (*p<0.1152*). The reported seasonal mean value is 1.07 mg/L, higher than the 2012 seasonal mean of 0.81 mg/L (*p<0.0091*).

Riverine C stocks and fluxes may change over time as plant succession generates higher pools of soil organic carbon (SOC) in the organic horizon and releases these pools seasonally. Export of these pools in regions of frozen soil as DOC (**Fig 2A**) is facilitated by seasonal weather patterns. During the spring freshet, winter-accumulated snow melts quickly and water runs overland where it dissolves the abundant, less-degraded carbon present in shallow soil layers before entering streams, releasing stored nutrients and flushing the terrestrial landscape. This process does not involve a deep penetration because soil is still frozen, which constitutes an impermeable barrier prior to thawing [30]. With an increase air temperature and thaw depth, subsurface flowpaths through soils migrate deeper and the surface water nutrient composition changes when soil organic C and porewater DOC is flushed. Leaching occurs laterally along seasonal frost or discontinuous permafrost boundaries until pooling or draining into a stream/river basin.

Water δ$^{18}$O at May Creek (**Fig 2B**) increased seasonally in 2012 (*p<0.0001*), from -22.68 ppm on May 14$^{th}$ to -21.99 ppm on August 20$^{th}$. For 2013, water isotope values did not increase seasonally and expressed a mean of -21.64 ppm. A similar trend was present for Young Creek, with 2012 values increasing from -23.09 ppm on May 14$^{th}$ to -22.69 ppm on August 20$^{th}$; the first two seasonal values in the month of May were high. No seasonal

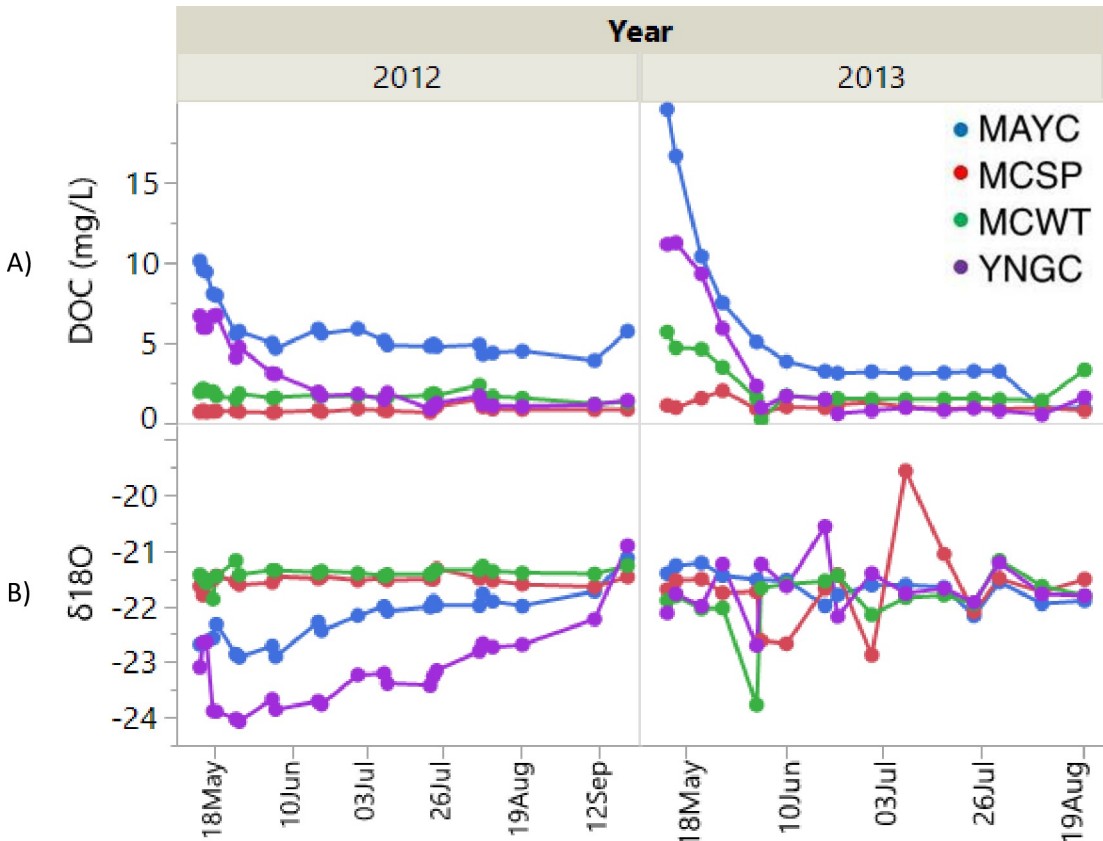

**Fig 2. Seasonal and interannual trends.** (A) DOC concentration and (B) water δ18O for Creek (MAYC and YNGC), subsurface fed spring (MCSP), and mixed source (MCWT) sites.

enrichment was observed in 2013; a mean value of -21.69 ppm is reported. Water source may be inferred from, among others, shifts in the $\delta^{18}O$ and $\delta^{2}H$ values for stream and river water [55–58]. Typically, when snow melt is the dominant source for river systems, $\delta^{18}O$ and $\delta^{2}H$ values are depleted compared to seasonal transitions into rainfall driven hydrographs that exhibit enriched $\delta^{18}O$ and $\delta^{2}H$ values, following the classic seasonality of precipitation isotopes [57, 59]. River water $\delta^{18}O$ values in our study region increased seasonally in 2012 due to a shift in source from snowmelt to an active layer thaw and precipitation event-driven hydrograph (**Fig 2B**). In 2013 however, this temporal pattern was not observed with surface water values being relatively consistent at the seasonal mean of -21.6 ‰ from May to September. This consistency in the surface water isotope values is likely the result of a rapid snowmelt and warm air temperatures from May-July (**S2 Fig**) preventing seasonally-thawed soil porewater from being sufficiently flushed. Our observations from 2012 indicate a snowpack with melt water input being more ephemeral and isotopically enriched rainfall becoming a larger fraction of river water sources as the growing season progressed.

## Hydrological characteristics and in-situ observations

Peak flow was observed between late May and early June, with annual maxima of 0.54 cms on June 12th (2012) and 0.63 cms on May 28th (2013). Discharge tapered through the end of August (**Fig 3A**), with minima of 0.13 cms for August 23rd (2012) and 0.08 cms for August 16th (2013). Discharge tended to occur in pulses and correlated strongly with precipitation

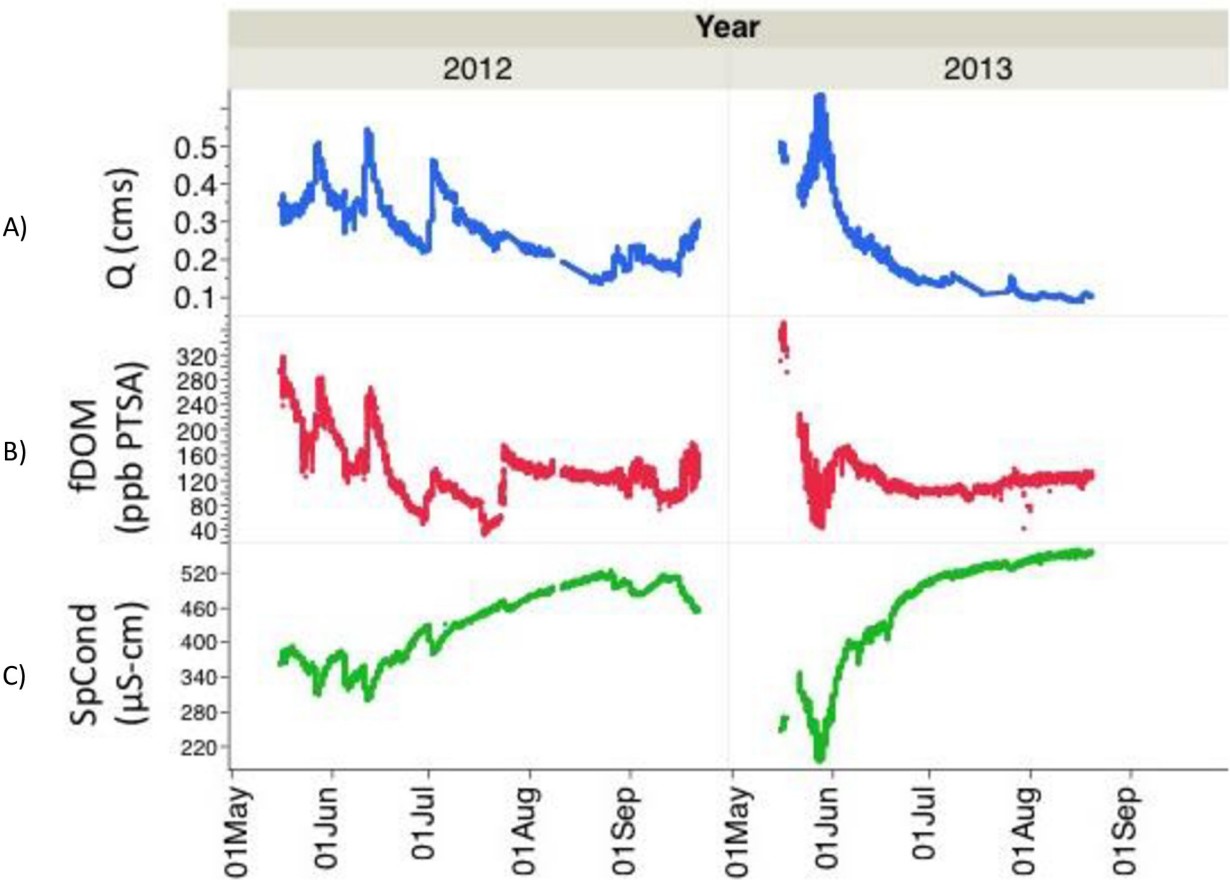

**Fig 3. Real-time *in-situ* hydrologic data collected from sensors deployed in May Creek over 2012 and 2013 at 30 cm depth.** Data are 5-minute averages of (A) discharge (in $m^3 s^{-1}$), (B) fDOM (in ppb PTSA standard), and (C) specific conductance (in μS-cm).

events in 2012. For 2013, the hydrograph was driven by one large pulse in late-May and with several minor pulses throughout the season. The major nutrient pulse for 2013 differed from others in 2012 and later in the season in that the pulse was accompanied by a decrease in specific conductance and fDOM (**Fig 3B**). Snow water equivalency reached 0 mm on April 28 and May 12 for the 2012 and 2013 water years, respectively (**S1 Fig**). Soil Temperature (5 cm) values recorded during 2013 revealed a seasonal increase from 0 °C on May 17[th] to a maximum of 12.5 °C on July 29[th].

Specific conductance (SpCond), a measurement of the electric current in the water carried by ionized substances, increased steadily throughout both seasons at May Creek (**Fig 3C**). We noted minima of 298 μS-cm on June 12[th] (2012) and 193 μS-cm on May 28[th] (2013). Maxima were observed at 523 μS-cm on August 26[th] (2012) and 559 μS-cm on August 16[th] (2013). The release of such ionized compounds were diluted during precipitation events and negatively correlated with discharge spikes.

Seasonal fDOM levels correlated with DOC values obtained from the grab samples (**Figs 2A and 3B**). Coupled with episodic pulsing events, fDOM values decreased seasonally both years and correlated strongly with discharge. For both seasons, fDOM values fluctuated diurnally with pronounced amplification in mid-May tapering off to low amplification in late July. For May 17[th], a representative spring freshet date, mean values of 281 ppb PTSA and 0.34 cms

for 2012 and 346 ppb PTSA and 0.48 cms for 2013 were observed. As the growing season progressed, both fDOM and discharge decreased. On July 29th, a representative summer date at maximum soil T, fDOM and discharge were observed at 144 ppb PTSA and 0.24 cms for 2012 and 116 ppb PTSA and 0.11 cms for 2013, respectively. These values correlated (both positively and negatively) with episodic precipitation events; For 2012, both fDOM and discharge increased following precipitation as nutrients stored in a relatively shallower snowpack were released following multiple rain events (S2 Fig). For 2013, these values negatively correlated during one large multi-day pulse event from May 23rd to June 2nd, when a deeper snowpack on top of a delayed snowmelt (S1 Fig) was observed. fDOM values at this time decreased in response to increasing discharge and decreasing specific conductance.

Seasons with early snowmelt, as in 2012, create highly efficient conditions to leach extractable DOC from soil mineral layers. In 2013, a combination of delayed, rapid snowmelt with an onset of low precipitation and warm air temperatures from May-July (S2 Fig) may have led to low soil moisture with the occurrence of small, isolated pockets of flow paths that allowed the infrequent precipitation to advect into streams without interacting with soil C. Our findings for 2013 support this hypothesis, since the major hydrograph pulse event involved a drop in fDOM values, an enrichment $\delta^{18}$O values, and a drop in SpCond. Seasonal flow path structure is known to have an impact on runoff rates and DOC export [60, 61]. Extreme events (i.e. outburst floods, slope failure from permafrost melt, precipitation pulses, etc.) play a critical role in stream ecosystems, and extreme values may be more important than mean values since the absence of continuously-recording instruments collecting such data would leave the detection of extreme events to chance. Consequently, what may matter most is to be able to measure changes in the frequencies and magnitudes of these events, rather than changes in mean values. Much of the variability observed in water chemistry parameters in the Wrangell-St. Elias National Park has been noted as pulse-dependent [62], and Fig 3 indicates for the first time this real-time interdependency between fDOM, discharge, and specific conductance. The fluorescent fraction of the CDOM pool, fDOM has been implicated recently as a proxy for DOC concentrations in the marine environment [63], and recent efforts [64–66] have been directed at determining high-resolution continuous DOC flux for freshwater systems. The fDOM data from this study reveals pulse events that can both increase and decrease DOC following precipitation, and that this is presumably dependent on snowmelt timing and seasonal flowpath structure.

## Seasonal fluorescence characteristics

May Creek FI values (Fig 4A) increased seasonally for both 2012 (*p<0.0071*) and 2013 (*p<0.0049*), from 1.54 on May 14th to 1.64 on Sept 21st (2012) and from 1.51 on May 14th to 1.65 on August 20th (2013). Humification Index (HIX) values at May Creek (Fig 4B) did not decrease seasonally, with mean seasonal values of 0.901 and 0.907 for 2012 and 2013. For Young Creek, the same seasonal trend was observed, with an increase in FI values for both 2012 (*p<0.0339*) and 2013 (*p<0.0002*) from 1.61 to 1.66 (2012) and from 1.56 to 1.77 (2013). HIX values at Young Creek did not decrease in 2012 (seasonal mean value 0.8427) but did decrease significantly in 2013 (*p<0.0004*) from 0.8994 on May 14th to 0.7456 on August 20th. Both creek systems exhibited lower FI values and higher HIX values than the sites receiving predominantly subsurface-derived water. MC Spring exhibited mean FI values of 1.79 and 1.83 for 2012 and 2013, with no seasonal increase (*p<0.6288 and p<0.5892* respectively*).* HIX values at MC Spring did not change seasonally (p<0.9530) in 2012, with a seasonal mean value of 0.5167. However, HIX did increase significantly in 2013 (*p<0.0116*) from 0.6140 to 0.6770. Thus, both creeks provide a mostly humic terrestrial contribution of DOC with a seasonally

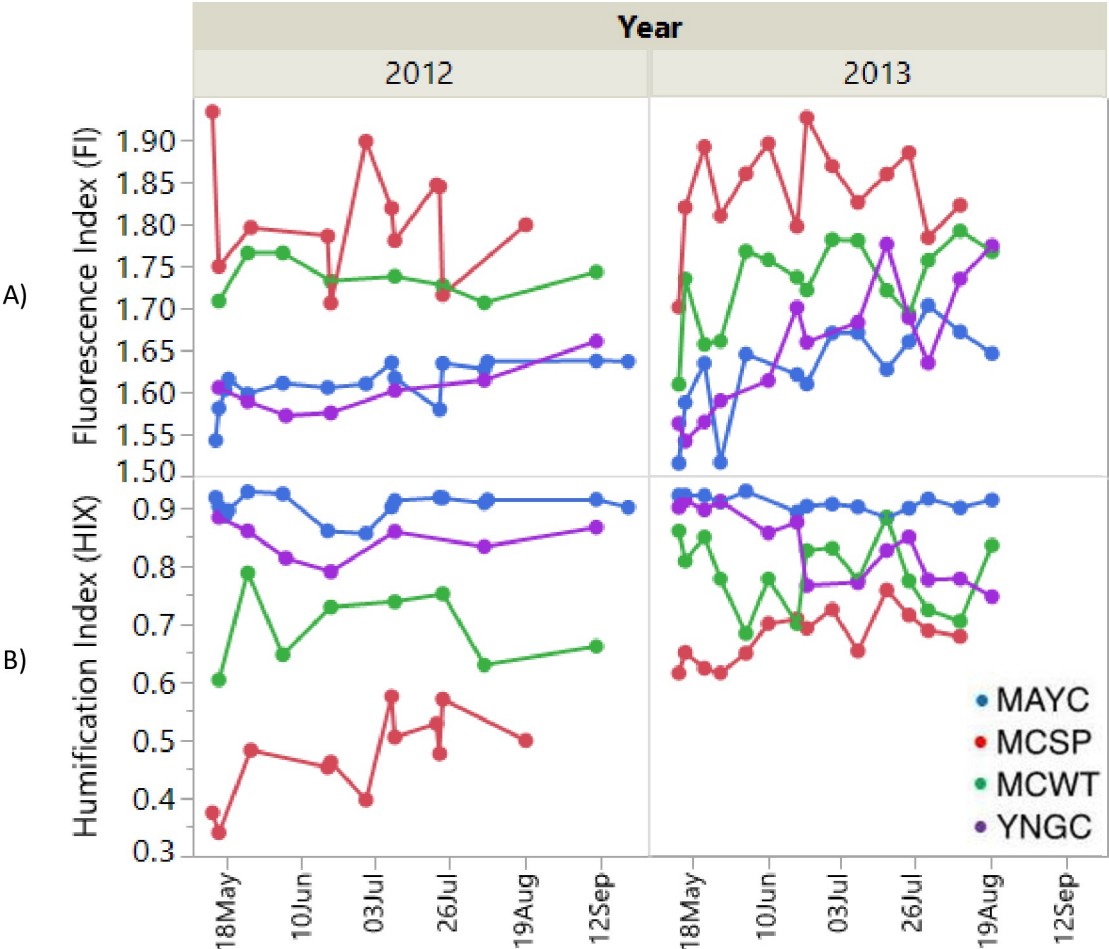

**Fig 4. Seasonal and interannual fluorescence characteristics by site for 2012 and 2013.** (A) Fluorescence index and (B) Humification index.

decreasing terrestrial contribution later in the fall, presumably due to an increasing contribution from groundwater. MC Spring exhibited consistently more biogenic DOC and less humified character than all other sites, with MC Wetland showing similar MC Spring-like character in 2012 and a mixture of sources in 2013.

The fluorescence characteristics (FI and HIX) of filtered water are commonly used to infer the relative composition of terrestrial vs. biogenic composition (FI) as well as the relative extent of highly aromatic, humified C to proteinaceous C (HIX). The seasonal values of each of these indices by site, as presented in **Fig 4**, suggest subsurface-dominated flowpath regions export DOC of a different origin from highly terrestrial-draining surface water sources. From May-September FI values increased at both May Creek and Young Creek, indicating that DOC becomes increasingly microbially-derived as opposed to terrestrially-derived [67]. This is consistent with the seasonality observed traits of the Yukon River Basin [27] and this DOC presumably becomes increasingly ancient, as inferred by [26]. DOC output in May Creek is more terrestrially-derived because microbial activity is low due to cold temperatures. The subsurface source MC Spring revealed significantly lower HIX values than May Creek and Young Creek, which indicates high H:C ratios as humification is not fully progressed [67] and derived from extracellular release and leachate from bacteria. MC Spring is also higher in ionic

strength as confirmed by specific conductance readings (**Fig 3C**). Interestingly, HIX values for Young Creek decreased seasonally in 2013, while MC Spring increased with values converging towards the end of the season. DOC values from 2013 season revealed a rapid mobilization of dissolved C stocks in May, with lessening snowmelt and shallow active layer contribution as summer progressed. Young Creek is a less forested and a larger non-glacial fed watershed compared to May Creek with steep banks eroding sites of discontinuous permafrost, which when thawed, pulse into the creek. Young Creek HIX values indicates DOC contributions are less-humified than May Creek and that they increase seasonally, a pattern which is not observed at the highly-forested May Creek. This is presumably due to the larger standing forest biomass and relative contribution to throughfall C in May Creek. Young Creek drains a higher elevation basin, much of which is above treeline. Although humic-like components still dominate the DOC pool in May and Young Creeks, a deepening active layer in a warmer boreal forest could potentially release additional biogenic-C in to the riverine system and potentially into the downstream rivers such as the Copper River and into the GoA with cascading consequences for in-stream biology and oceanic food webs.

We obtained a validated 3-component PARAFAC model for the EEMs data from both sampling years (**Fig 5A**). C1 is generally referred to as Peaks A and C and are described as regions found in freshwater ecosystems and contain fluorescence signatures related to highly degraded, aromatic DOM [68–71]. C2 is described in the literature as Peak M, or microbial derived humic peaks, which are comprised of relatively aliphatic and low molecular weight DOM [68, 69]. C3 region is generally described as tyrosine-like fluorescence [72, 73]. Plotting HIX vs C1 (**Fig 5B**) shows that as HIX increases, so does the relative contribution of C3, which makes sense given that higher HIX values describe compounds that highly aromatic with high oxygen, hence the longer emission wavelengths. Plotting HIX vs C3 (**Fig 5B**) shows that as HIX decreases, the relative contribution of C3 increases. Again, this makes sense given that low HIX values are associated with newly formed compounds that are highly aliphatic and contain low oxygen, hence the shorter wavelengths. The strong positive correlation with HIX and C3 from May to August for both years suggests that aliphatic compound are produced at the end of the season. Similar fluorescence signatures found from earlier studies (similarity score > 0.98 in OpenFluor), overlapped with identified components from this study [74]. Component 1 matched fluorescence signatures from 26 studies, C2 3 studies [75–77] and C3, 2 studies [78, 79]. Most notable were the spectral properties of component 1 at ex max 255–260 nm and em max at 448–480 nm are humic-like signatures found in forested systems and matched C2 from [80, 67]. Component 2 with spectral properties of Ex325/Em396 nm was compared to C6 from [81], C2 from [75] and C2 from [76] to an extent. This fluorescence signature is often referred to as the "M Peak" and is ubiquitous in a wide range of environments [68, 69]. Component 3, referred to as protein-like fluorescence at ex max 270–275 nm and emission 304–312 nm, overlapped with spectral signatures of C3 from [78] and C3 from [79]. Comparing the spectral properties with other studies in boreal forested systems, this component matched protein-like fluorescence component C13 from [80]. Consistent with [26, 32, 82, 83], we identify the fluorescence components in this watershed as phenols, lignin, tannins, gallic acid and amino acid tyrosine.

Our EEMs exhibited excitation/emission fluorescence in regions corresponding to UVA (ex 290–325 nm, em 370–430 nm) and UVC (ex 320–360 nm, em 420–460 nm) noted in [67]. Using the strong correlation and inverse correlation of C1 and C3 respectively to HIX (**Fig 5B**) combined with seasonal trends shown in **Fig 4**, we can identify that May Creek and Young Creek were dominated by this fluorescence character, with Young Creek showing decreased humification by August and September. MC Spring however, displayed a protein-like signal (tyrosine-like ex 270–275 nm, em 304–312 or tryptophan-like ex 270–280 nm, em 330–368

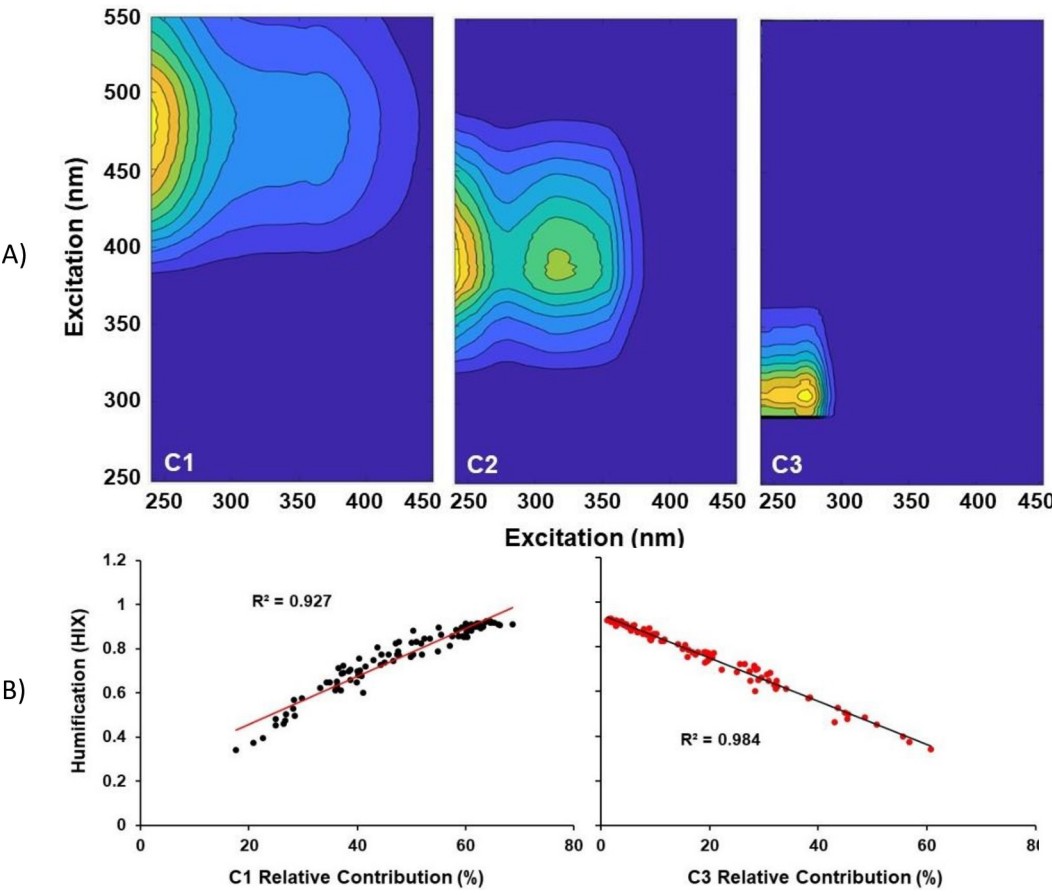

**Fig 5. Parallel Factorial Analysis (PARAFAC) of fluorescence dataset.** (A) PARAFAC loadings for a validated 3-component model. (B) Strong correlations to humification index (HIX) are presented.

nm) throughout the season, with an increase in UVA/UVC DOC character in fall. The mixed source MC Wetland exhibited MC Spring-like (low humification, high FI) DOC in 2012, but was creek-like (high humification, lower FI) in 2013. As also purported by FI and HIX indices, Young creek exported less UVA and UVC-like character in late summer, due to less relative tree cover and more contribution from mobilized thawing processes. As air temperatures and thus thaw depth increase, ionized chemical species associated with specific conductance (**Fig 3C**) and of biogenic origin (**Fig 4A**) sequestered at lower depths are mobilized following precipitation events and subsurface advection. With warming temperatures and a deepening active layer, we would expect an increased export resembling this chemical character.

## Net ecosystem exchange, $CO_2$ flux, and correlations to DOC

NEE values for this boreal forest show consistency in the diurnal patterns being a net source of $CO_2$ to the atmosphere during the nighttime periods, and a strong sink during mid-day, regardless of month (**S3 Fig**). The magnitudes of nighttime respiration changed from 2.2 µmol m$^{-2}$ s$^{-1}$ in May 2012 to 6.0 µmol m$^{-2}$ s$^{-1}$ in July. The mid-day peaks of NEE are lowest in autumn (September) at -4.1 µmol m$^{-2}$ s$^{-1}$ and highest in July at -11.9 µmol m$^{-2}$ s$^{-1}$. The sharp diurnal switch from source to sink of the monthly averages also reflect the seasonal transition from shorter to longer days at the higher latitudes where by the transition becomes very abrupt

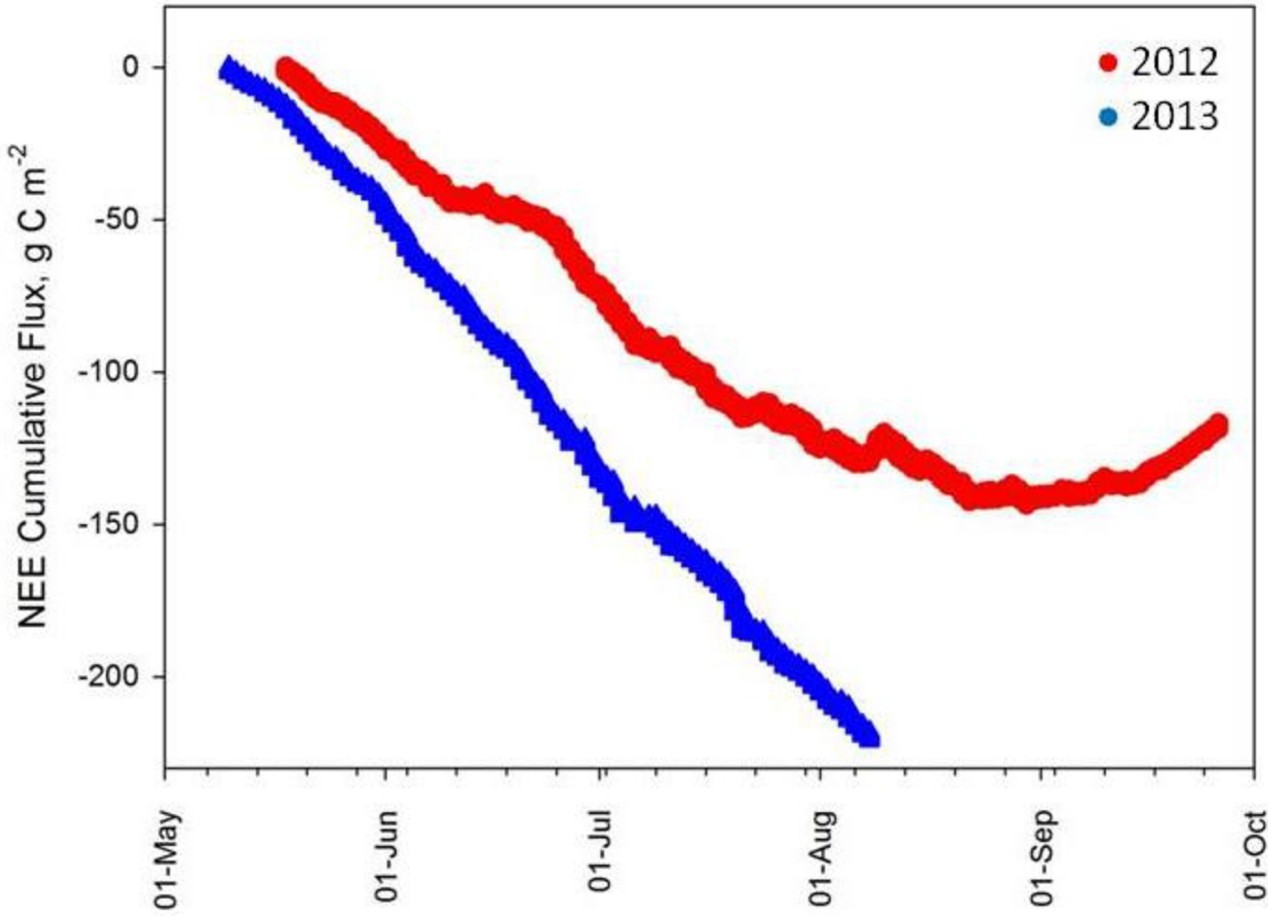

**Fig 6. Net ecosystem exchange of may creek watershed for 2012 and 2013.** Data are half-hourly averages, expressed as g C/m$^2$, collected from eddy covariance flux tower installed on-site.

in June and July near the summer solstice and occurs hours earlier than in May and again in September.

During the two years of study, the rates and magnitudes of C accumulation (**Fig 6**), were lowest and slowest in 2012, with the maximum amount in 2012 being 142 g C m$^{-2}$. In 2013 however, the C accumulation rate began earlier and never plateaued during the measurement period reaching a maximum of 220 g C m$^{-2}$. This supports the hypothesis that atmospherically-fixed C is accumulating in the standing biomass and accounts for the majority of the total C budget in headwater streams in this south central Alaskan system. Taking mean annual DOC and discharge values into account, our data indicates DOC export for May Creek from May to September to be 0.57 g C/m$^2$ for 2012 and 0.40 g C/m$^2$ for 2013. These values compare with other late successional catchment-derived C of 1.64 g C/m$^2$ for the Yukon River Basin [84], 1.5–8.3 g C/m$^2$ for a bog of Ontario, Canada [85] and 4.6–17.6 g C/m$^2$ for a central Siberia site underlain with continuous permafrost [86]. According to [84], DOC export values vary considerably by catchment properties, and large values in excess of 20 g C/m$^2$ yr are typically seen when precipitation greatly exceeds evapotranspiration and/or upland area exceeds wetland area. Conversely, DOC export is low, < 10 g C/m$^2$ yr, when catchment is dominated by peatland, relief is negligible, and/or precipitation and evapotranspiration values are similar

[84, 85]. In May Creek, it is believed that low relief and dense standing biomass contribute to the low values compared with other studies.

$CO_2$ exchange in the May Creek Watershed exhibited diurnal and seasonal patterns that are consistent with ecosystems globally and in forests throughout the northern hemisphere [24, 87, 88]. Irradiance levels fluctuate diurnally from being zero for up to 8 hrs per day to maximum values of $>1500$ µmol m$^{-2}$ s$^{-1}$ during mid-day; driving in large part the variation in GPP and thus NEE we observe daily as reported for other boreal and temperate forests [87–89]. Carbon sequestration over the course of the growing season we observed are within the ranges of other North American boreal forests, including the sites of the BOREAS program and other Canadian boreal sites [87, 90, 91]. However, none of these programs have placed C sequestration in the context of both the gaseous and aqueous exchanges of C in a headwater stream; in our system the gaseous exchange dominates the budget. However, if boreal watersheds are less forested and more dominated by bogs and fens, the relative contribution of gaseous to aqueous C cycling maybe significantly different than what we observed for May Creek, in the interior of south central Alaska [91]. C fixation may be lower, but DOC efflux may also be lower.

Understanding the magnitude of DOC export from boreal watersheds in the context of the watershed C net $CO_2$ exchange is paramount to recognizing the scope and scale of basin-wide C biogeochemical cycling; but seldom is this accomplished as it requires an interdisciplinary approach and a holistic perspective. Our results indicate that this boreal forest ecosystem is exceedingly C conservative during the growing season and acting as a very strong overall C sink. Seasonal weather patterns appear to be a greater driver of the observed variability in DOC quantity and composition than does cumulative C fixation, with DOC export constituting less than 5% of summer NEE. This exceedingly tight C cycle is reflective of the storage of C in standing biomass, in forest understory species and in the formation of soil C pools that are relatively recalcitrant. This strong C-sink capacity of these watersheds are however venerable to catastrophic events that may alter DOC export and C fixation such as fires, which are know to very rapidly oxidize forest biomass, remove understory protection of soil C pools and result in large C source traits over short and longer time periods until successional processes restore the forest ecosystem and the thermal stability of the underlying soil c pools [92, 93].

## Conclusion

In this study, we examined the seasonal hydrologic and atmospheric processes that control C exports and storage in a late-successional forest ecosystem and anticipate such mobilization in currently glaciated areas of the basin to be similar to May Creek in the coming years as the significant glacial coverage (18% of the basin, [94] melts, permafrost thaws, and vegetation advances). Pulse events and seasonal weather patterns in this region have a major effect on the timing, magnitude, and chemical characteristics of exported DOC, and understanding these effects is paramount to predicting future C flux.

## Supporting information

**S1 Fig. Snow water equivalent (SWE), Precipitation accumulation, and air temperature data for periods of spring freshet.** Data are expressed for (A) 2012 and (B) 2013 field seasons. Reproduced from SNOTEL site 1096: May Creek.
(TIF)

**S2 Fig. Seasonal cumulative precipitation, snow water equivalent, and air temperature values for May Creek.** Data reproduced from SNOTEL site 1096: May Creek.
(TIF)

**S3 Fig. Diurnal $CO_2$ flux (µmol m-2 s-1 $CO_2$) characteristics at May Creek for 2012 field season.**
(TIF)

**S1 File. May creek basin grab sample data.**
(XLSX)

**S2 File. May creek sensor data.**
(XLSX)

**S3 File. May creek basin fluorescence data.**
(XLSX)

**S4 File. May creek net ecosystem exchange data.**
(XLSX)

**S5 File. May creek snotel data.**
(XLSX)

## Acknowledgments

We thank the community members of McCarthy, Alaska for their immeasurable contribution of knowledge to inform our research and for providing logistical support in the field. We thank Eric Veach at the National Park Service for his guidance with permitting and compliance with eddy tower construction. Special thanks to John Ferguson for database support, Aaron Dotson and Jason Burkhead (UAA) for providing fluorometer access *gratis*, Matt Rogers at UAA-Stable Isotope Facility for technical assistance, Isaac Hayes, and Nathan Welker for field work assistance. Data used for this study are available in the Figs and may be obtained by contacting the corresponding author.

## Author Contributions

**Conceptualization:** Patrick L. Tomco, Rommel C. Zulueta, Leland C. Miller, Jeffrey M. Welker.

**Data curation:** Patrick L. Tomco, Rommel C. Zulueta, Leland C. Miller, Phoebe A. Zito.

**Formal analysis:** Patrick L. Tomco, Rommel C. Zulueta, Leland C. Miller, Phoebe A. Zito.

**Funding acquisition:** Robert W. Campbell, Jeffrey M. Welker.

**Investigation:** Patrick L. Tomco, Rommel C. Zulueta, Jeffrey M. Welker.

**Methodology:** Patrick L. Tomco, Rommel C. Zulueta, Leland C. Miller, Jeffrey M. Welker.

**Project administration:** Robert W. Campbell, Jeffrey M. Welker.

**Resources:** Rommel C. Zulueta, Leland C. Miller, Robert W. Campbell, Jeffrey M. Welker.

**Software:** Patrick L. Tomco, Rommel C. Zulueta, Phoebe A. Zito.

**Supervision:** Rommel C. Zulueta, Jeffrey M. Welker.

**Validation:** Rommel C. Zulueta, Jeffrey M. Welker.

**Visualization:** Rommel C. Zulueta, Phoebe A. Zito, Jeffrey M. Welker.

**Writing – original draft:** Patrick L. Tomco.

**Writing – review & editing:** Patrick L. Tomco, Phoebe A. Zito, Robert W. Campbell, Jeffrey M. Welker.

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
