## [Decision Letter · Decision Letter 0]

10 Sep 2019

PONE-D-19-14655

Pulse driven DOC inputs control the C exports in May Creek: a late-successional headwater boreal forest watershed of the Copper River Basin, Alaska

PLOS ONE

Dear Dr. Tomco,

Thank you for submitting your manuscript to PLOS ONE. After careful consideration, we feel that it has merit but does not fully meet PLOS ONE’s publication criteria as it currently stands. Therefore, we invite you to submit a revised version of the manuscript that addresses the points raised during the review process.

We would appreciate receiving your revised manuscript by 11 October 2019.  To enhance the reproducibility of your results, we recommend that if applicable you deposit your laboratory protocols in protocols.io, where a protocol can be assigned its own identifier (DOI) such that it can be cited independently in the future. For instructions see: http://journals.plos.org/plosone/s/submission-guidelines#loc-laboratory-protocols

We look forward to receiving your revised manuscript.

Kind regards,

Robert L. Bradley

Academic Editor

PLOS ONE

Journal Requirements:

2. Please include in the Methods section the names of the locations, and provide geographic coordinates for the data set.

3. We note that Figure  [1] in your submission contain [map/satellite] images which may be copyrighted. All PLOS content is published under the Creative Commons Attribution License (CC BY 4.0), which means that the manuscript, images, and Supporting Information files will be freely available online, and any third party is permitted to access, download, copy, distribute, and use these materials in any way, even commercially, with proper attribution. For these reasons, we cannot publish previously copyrighted maps or satellite images created using proprietary data, such as Google software (Google Maps, Street View, and Earth). For more information, see our copyright guidelines: http://journals.plos.org/plosone/s/licenses-and-copyright.

You may seek permission from the original copyright holder of Figure [1] to publish the content specifically under the CC BY 4.0 license. 

If you are unable to obtain permission from the original copyright holder to publish these figures under the CC BY 4.0 license or if the copyright holder’s requirements are incompatible with the CC BY 4.0 license, please either i) remove the figure or ii) supply a replacement figure that complies with the CC BY 4.0 license. Please check copyright information on all replacement figures and update the figure caption with source information. If applicable, please specify in the figure caption text when a figure is similar but not identical to the original image and is therefore for illustrative purposes only.

Reviewers' comments:

Reviewer's Responses to Questions

**Comments to the Author**

1. Is the manuscript technically sound, and do the data support the conclusions?

Reviewer #1: Partly

Reviewer #2: Yes

2. Has the statistical analysis been performed appropriately and rigorously? 

Reviewer #1: I Don't Know

Reviewer #2: I Don't Know

3. Have the authors made all data underlying the findings in their manuscript fully available?

Reviewer #1: Yes

Reviewer #2: Yes

4. Is the manuscript presented in an intelligible fashion and written in standard English?

Reviewer #1: Yes

Reviewer #2: No

5. Review Comments to the Author

Reviewer #1: GENERAL COMMENTS:

The manuscript provides a useful carbon dataset that helps link forest/overland carbon flux with stream export of dissolved organic carbon (DOC). These data and findings will be of interest to a broad audience working on global to boreal carbon cycles. The authors present an interesting story around seasonal drivers of carbon flux and help highlight how growing condition (light and temperature) and water (precipitation and snowmelt) influence differently dissolved and gaseous carbon flux. The manuscript has many positives and I think a few modifications are needed to help the manuscript more fully explore the data and bring together the full story. I have three main suggestions that I think would make the manuscript more useful.

1) Make full use of the EEMs --- Given that you have the full EEM, why only focus on FI and HIX. Peak picking (Paulo Coble's work) or PARAFAC might provide more useful insight when tested through a multivariate DOM method. SUVA and spectral slope would also be useful measure to understand how the complexity of DOM changes across the growing season. The discussion starts to exam the EEMs visually, but I think it would be useful to take the analysis farther by using the full set of DOM parameters that optical chemistry can generate. If these data are then analyzed through multivariate analysis, one can speak about the DOM as a pool and detailed univariate explanations of all the variables could be avoided.

2) Statistics, questions, and hypothesis tested needs more detail and justification --- It is unclear to me what test and question were used to generate p-values in this study. More detail is needed for the reader to understand what data are being compared, why, and how. For example, how did you define season? How are you testing rate of change across time? I think better defining the studies hypothesis and question with the link statistics will make the results easier to follow with respect to your broader story.

3) Combining results and discussion sections will make it easier for your reader to follow the data and the story --- Overall the results are written clearly and as individual statements I understood the pattern and result being described. I think what I struggled to understand was how all the variables and different temporal scales that where tested fit together as one story. The discussion and abstract did a fantastic job describing the overall story but at times I struggled to link the broad story to the specific results. I think to help the reader move through your complex data set, combining the results and discussion section would communicate more clearly the big picture story and the result that supports that big idea. This might shorten then manuscript a little as well better highlight the interpretation of the result alongside the result.

SPECIFIC COMMENTS

Abstract: The abstract is clear and provides a nice snapshot of the study and its relevance. I did not fully understand the last sentence of the abstract. I am not certain what shift is being referenced and I am not certain in what respect this carbon mass is integrated and important. Could the sentences be revised to be more exact?

Key Points:

1) Does this include the frozen period? It seems odd that CO2 fixation wouldn't change between growing and dormant seasons. This point might need to be qualified with the timer period of the study

3) Browning has multiple means. Since your paper is about DOC, could you use another word to describe the drought impact on the forest?

4) Did you mean carbon or DOC rather than nutrient?

Introduction: the introduction is framed well and does a nice job setting up the study and the need for the combined study of carbon gas and DOC flux. The introduction was a little long. You could likely remove the paragraph about DOM fluorescence without loss to your overall story. To me this approach is now very common and it's OK to not focus on how DOM was characterized but focus instead on what knowledge around the DOM characteristics can tell us about C cycles in this study

L150-L155: I think MC needs to be define in text. MC could also be used when describing the actually creek in text as it was in the figures

L189: Did you mean fluorescence rather than absorbance?

L338-L339: The July value is nearly 3x higher. This seems a much larger change in magnitude than the water isotopes and fDOM, FI, and HIX values above, which were interpreted as significantly different. Why is this considered stable?

L444: This should be a statistical inspection using multivariate methods to show how the DOM pool changes. These visuals help show the pattern, but I think the data need to be analyzed statistically, in order to show the magnitude and direction of change.

L459-L462: This hypothesis needs to be stated in the methods with statistical test explaining how the hypothesis was tested

Figure 5: Could error bars be added to each point to show how much day to day or night to night variation is displayed in each monthly mean at each time point?

Figure 7: These are useful plots. I think the data within the EEMs needs to be explored in the results section through PARAFAC or peak picking. Otherwise, this rich data set isn't explored to its fullest. In terms of the plot, the xyz axis font is to small and not a very good resolution. I zoomed in and the text was fuzzy and difficult to read. You should make a note that the z-scales vary from plot to plot and specify the units (RU, QSU, etc...). Conversely, you could standardize each plot by the max value and this would allow all plots to be displayed on the same relative scale. The second option would draw the reader to changes in characteristics of DOM between spring and fall rather than highlight changes "concentration"

Reviewer #2: Review PlosOne.

This manuscript presents the results of a two-year study conducted in a boreal watershed of Alaska. The authors describe the fluxes (magnitude, seasonal and annual variations) and chemical characteristics of DOC in four stream types of watersheds characterized by different vegetation cover and soil characteristics. They also measured CO2 fluxes for one growing season at one of the sites to assess the significance of DOC exportations as compared to C fixation through photosynthesis. This study was motivated by the fact that global warming may increase DOC outputs and reduce CO2 fixation in these ecosystems, which may turn them into C sources. This is an interesting and important topic that researchers must tackle. The main conclusion is that the watershed does not export large amounts of DOC as compared to C fixation. Therefore, in contrast with their hypothesis the watershed is still a C sink and highly C conservative. They also found that most DOC efflux occurred as a pulse during snow melt in May.

The amount of work, the sampling effort (four streams), the methods (eddy covariance, fluorescence spectroscopy) and analyses (stable isotopes, DOC concentrations) deployed for this study are considerable. The authors described the materials and methods thoroughly and clearly, and the bibliography is thorough.

The results are interesting and deserve publication. However, I was disappointed by the way the authors presented, interpreted and discussed their results. First, the figures could be of better quality. I think for instance that the use of dots instead of line prevents from visualizing the seasonal trends. The resolution of Fig 7 is too low for the x- and y-labs to be readable. I think that the figures should also be reorganized to facilitate the discussion. For instance, why not grouping results with similar trends in a single figure (e.g., fDOM with DOC flux, and delta 18O with Q)?.

The authors insist on the importance of precipitation and temperature on DOC fluxes in this type of ecosystem but did not include the data with the ms and didn't discuss their results properly. I would bring some of these data from the supplementary materials to the main text and maybe remove the Fig 5 which does not provide particularly insightful information. The fact that the vegetation is a source of CO2 at night and a sink during the day is quite obvious.

In addition, I found the discussion section not very well written and organized. I think the interpretation of the results could be clearer and more convincing. They, in my opinion, do not do a satisfactory job at answering basic questions that come to mind after reading the results section (see in comments below).

The abstract as well as the introduction describe the study sites and the context thoroughly, but the abstract does not emphasize the results and the main conclusions of their work, which in my opinion should be clarified. I also think that the results could have been discussed more thoroughly and in a clearer way. For instance, the fDOM results are barely discussed.

Here are my comments for each section of the manuscript.

Abstract:

The first paragraph could be shortened (in my opinion, only the first and last sentences should be conserved) and the authors should put more emphasis on their results and conclusions. Several sentences are very long and could be split to clarify the message.

The writing could be improved.

Some statements are not entirely supported by the results or not explained well enough.

L. 18-21: Both aqueous and gaseous C processes are key drivers of food webs and climate feedbacks in these ecosystems, regardless permafrost or glaciers are melting or not. This sentence is clumsy. The second part of the sentence (“as northern…”) is redundant with the first part. It is excessively long for the message it conveys.

L. 30. DOC “inputs” or “outputs”?

L. 31-35. This sentence is too long. Split before “and fluorescence…”

L. 31. “depended on seasonal subsurface flowpath structure”. Not very clear. What do you mean exactly?

L. 35-38. “DOC fluxes…”. This sentence is too long and not clear. If there is a positive relationship between NEE and DOC outputs state it clearly. “revealing” is not appropriate. The first part of the sentence does not reveal anything like that. Of course the C that is fixed by the vegetation is internally processed. How couldn’t it be?

L. 40-42. Not supported by the data.

Introduction

Overall, I don’t think the introduction is very well organized and written. Some sentences are too long (be careful to not overuse “as” - e.g., L.79-80) and ambiguous.

There should be a logical flow between the paragraphs, which is not always the case here.

The authors should insist on the importance of the study on a climate change context.

L. 61-62. No capital letters for boreal forest. I don’t really like the way you use “most important” to compare biomes. What about “the boreal forest plays a large role in global C cycle due to …”

L. 64. Planet’s

L. 66. All ecosystems have gaseous and aqueous fluxes. Boreal forests are characterized by large DOC fluxes. Rewrite the sentence.

L.71. “for” instead of “in”.

L. 80. “increasingly”

L.81. Why “decreased river flows”? Precipitation is not expected to decrease in the area and permafrost and glaciers thaw may increase river flow. I may have missed something. Clarify.

L. 91-93. This sentence is not clear. I think you should describe the consequences of the previous sentences.

L. 94-95. This sentence is ambiguous. Do you mean that terrestrial vegetation provides stream water with C directly (without passing through the soil)? What is the substrate you’re talking about? SOC or plant litter?

L. 96. Litter inputs to what? Rivers or SOC?

L. 98. “to” DOC

L. 102. Why are gaseous C fluxes particularly more complex than in other ecosystems? Because of CH4 emissions due to anoxic conditions in the soil? explain.

L. 117. It partitions…

L. 120. Briefly explain the nature and origin of fulvic acids; that FA differ depending on their origin and biogeochemical processes; that FA coming from plant litter and the soil have a higher degree of aromaticity as compared to microbially-derived FA; that we can estimate the relative contributions of FA sources by measuring FI. Some of this info is in your M&M section.

L. 127 and L. 132. Dissolved “organic” C

Materials and methods

Sampling was well performed. This part is clear and well described.

A short explanation of the meaning of fDOM values or why they performed this type of analysis could be added.

What kind of stats was performed on DOC seasonal trends? Mann-Kendall tests are commonly used.

Results

Fig 2 and 4. These figures are not very clear. I would rather use lines than dots. It would help visualizing seasonal trends.

Fig. 2B. Why isn’t delta 18O expressed in per mil?

Fig. 7. The x- and y-axes are not readable

Why is there no title for the first paragraph?

Results from MCWT are not presented in the text.

L. 277. The first two d18O values of May are high at YC. Mention it.

L. 283. Why cms and not m3 s-1?

L. 285. It would be insightful to display precipitation and temperature data to verify whether stream flow correlated with precipitation and/or snow melt resulting from high temperatures.

L. 298. Negatively correlated.

L. 299. The trends are indeed similar, i.e. decrease from May to June, but the correlation is not obvious.

L. 301. I agree that it correlates well in 2012 but not in 2013. It seems that there is a time lag.

L. 301-302. The diurnally variation in fDOM is not visible from this figure.

L. 307-308. When does it correlated positively and negatively?

L. 316-317. FI values are always >1.5. which -according to what you write in the M&M section- means that all DOC is from microbial origin.

L. 328. Does “this” refer to “the combination of low FI and HIX”?

L. 337. “a” strong “sink”

L. 338. “changed”

L. 341. There are additional spaces after “at” and “s-1”.

L. 341-342. Where are the monthly averages?

Discussion

The section “seasonality of DOC export” is poorly written and not clear. The interpretation of the results is weak and no references are cited. The links between the different analyses (e.g. DOC, d18O, FI, HIX and conductance) should be discussed more deeply. One way to do that could be a reduction of the number of sections. The questions that come to mind after reading the results section are: Why is there a DOC concentration peak in May? Why is it higher at MAYC than at YNGC? Where does this DOC come from? Why is fDOM positively correlated with DOC flux? Why is fDOM positively correlated with Q in 2012 and negatively in 2013? Etc. The discussion should clearly answer these questions.

L. 353. “North” America. The first sentence could be split after “North America”.

L. 357. What do you mean by “patterns”?

L. 364. Riverine C what? Stocks? Fluxes?

L. 367. Is the amount of nutrients stored in the snow cover significant? What kind of nutrients are you talking about? Please, clarify this point.

L. 369. What about removing “however” and replace it by “which constitutes an impermeable …”. Do you mean that most DOC leached in May (during snow melt) comes entirely from the snow pack? Where does this DOC come from? Develop.

L. 371-373. This sentence is too long and contains too many “as”.

L. 374. What are these nutrients with high specific conductance?

L. 375. The influence of precipitation events and subsurface mechanisms you’re talking about here is not clear.

L. 386-390. The cause of the absence of trend in d18O in 2013 is not clear.

L. 393-394. Explain why. On average, the DOC export seems however as high in 2013 as in 2012.

L. 398. The high Q in May 2013 is due to a rapid snow melt. It was associated with low conductivity, low d18O and fDOM values, suggesting that DOC comes almost entirely from the snow pack. Can you explain why the snow pack contains so much DOC and where it comes from?

L. 420. You could provide more information regarding biogenic and terrestrial DOC.

L. 424. Explain that the DOC output in May is more terrestrially-derived because microbial activity is low due to low temperatures. The DOC that’s released from the snow pack is not recycled by soil microorganisms.

L. 437. Why does the throughfall C result in more humified DOC?

L. 441-443. I agree with this statement but it is not clear how your data support it. Develop.

L. 444-455. This paragraph is very descriptive. It should be moved to the results section. As mentioned earlier, the resolution of the figure is too low for the x and y labels to be readable.

L. 469. Your values are therefore very low as compared to those reported in the literature. How do you explain that?

L. 481-484. C fixation would be indeed lower but as you mentioned before (L. 470), the DOC efflux would also be lower.

L. 485-498. The last paragraph is good. In my opinion, the conclusion mentioned L.488, i.e. the watershed is C conservative, it is the main take home message and should appear in the title. The high seasonality of nutrients fluxes in boreal catchment is not new.

Conclusion

The conclusion is good but these points were in my opinion not developed enough in the discussion.

6. PLOS authors have the option to publish the peer review history of their article (what does this mean?). If published, this will include your full peer review and any attached files.

Reviewer #1: Yes: Clayton J Williams

Reviewer #2: No

---

## [Editor Report · Decision Letter 1]

1 Nov 2019

DOC export is exceeded by C fixation in May Creek: a late-successional watershed of the Copper River Basin, Alaska

PONE-D-19-14655R1

Dear Dr. Tomco,

We are pleased to inform you that your manuscript has been judged scientifically suitable for publication and will be formally accepted for publication once it complies with all outstanding technical requirements.

With kind regards,

Robert L. Bradley

Academic Editor

PLOS ONE
---

## [Editor Report · Acceptance letter]

12 Nov 2019

PONE-D-19-14655R1 

DOC export is exceeded by C fixation in May Creek: a late-successional watershed of the Copper River Basin, Alaska 

Dear Dr. Tomco:

I am pleased to inform you that your manuscript has been deemed suitable for publication in PLOS ONE. Congratulations! Your manuscript is now with our production department. 

With kind regards,

on behalf of

Dr. Robert L. Bradley 

Academic Editor

PLOS ONE